# Development of a Sensitive, Easy and High-Throughput Compliant Protocol for Maize and Soybean DNA Extraction and Quantitation Using a Plant-Specific Universal Taqman Minor Groove Binder Probe

**DOI:** 10.3390/genes14091797

**Published:** 2023-09-14

**Authors:** Roberto Ambra, Marco Marcelli, Fabio D’Orso

**Affiliations:** 1Council for Agricultural Research and Economics, Research Centre for Food and Nutrition (CREA-AN), 00178 Rome, Italy; 2Volta Institute, MIUR (Italian Ministry of Education, University and Research), 09036 Guspini, Italy; marcomarcel@tiscali.it; 3Council for Agricultural Research and Economics, Research Centre for Genomics and Bioinformatics (CREA-GB), 00178 Rome, Italy; fabio.dorso@crea.gov.it

**Keywords:** plant DNA extraction, silica, chaotropic salts, *Zea mays*, *Glycine max*, ribosomal 18S rRNA gene, real-time PCR

## Abstract

We report the optimization of a high-throughput, compliant DNA extraction method that uses standard format 96-well plates and a commercial automated DNA purification system (ABI PRISM^®^ 6100 Nucleic Acid PrepStation). The procedure was set up for maize and soybean, the most common GMO crops and the main ingredients of several foodstuffs, and compared with an EU-validated CTAB-based method. Optimization of the DNA extraction was achieved by applying self-prepared buffers (for DNA extraction, binding, and washing) on the PrepStation loaded with proprietary glass-fiber-coated purification plates. Quantification of extracted DNA was performed by real-time PCR using previously reported endogenous soybean lectin and maize starch synthase genes and a novel plant-specific universal TaqMan MGB probe that targets the 18S rRNA multiple copy gene. Using serial dilutions of both maize and soybean genomic DNAs, we show low PCR sensitivity and efficiency for the official TransPrep DNA extraction protocol compared to the CTAB-based one. On the other hand, using serial dilutions of a standard reference plasmid containing a 137 bp sequence cloned from the 18S rRNA plant-specific ribosomal gene, we demonstrate the high PCR sensitivity and efficiency of the optimized DNA extraction protocol setup with self-prepared buffers. The limits of detection and quantification of the 18S rDNA reiteration were consistent with the calculated values, supporting the suitability of the DNA extraction procedure for high-throughput analyses of large populations and small amounts of tissue.

## 1. Introduction

DNA extraction is a major bottleneck step in plant PCR analyses, especially in GMO, marker-assisted selection (MAS), and next-generation sequencing (NGS), where many samples have to be analyzed. The achievement of PCR-suitable DNA from plants is complicated because of tissue heterogeneity and complexity and the considerable presence of carbohydrates and phenolic compounds [1,2]. Low PCR sensitivity has been reported to be the result of the presence of inhibitor molecules that promote DNA degradation or affect enzyme activity [3]. CTAB was introduced for its ability to efficiently remove carbohydrates and other PCR inhibitors and to provide high-quality DNA, both in terms of purity and integrity [4], and protocols were used in validation trials in order to detect GMOs using PCR [5] and adapted to several raw and processed food matrices [6,7].

However, CTAB extraction has the disadvantage of using harmful reagents, such as phenol and chloroform, that are necessary in order to achieve the efficient removal of contaminants and inhibition of nuclease activity and require complicated and time-consuming purification procedures that have demanded several modifications [8]. For such reasons, silica-based nucleic acid extraction methods have been introduced since the late 1980s [9]. Such approaches rely on the binding of DNA to silica materials (such as glass, silica particles, or glass microfibers prepared from diatomaceous earth) in the presence of a high concentration of chaotropic salts in order to enrich DNA and remove contaminant molecules from several organic and inorganic matrices. Subsequent reports have demonstrated that they are also appropriate for downstream applications like PCR [10], including forensic analysis [11]. Commercial kits for automated DNA extraction are available, however, because of their cost downside, starting from starting in the 2000s some studies have addressed the possibility to set up artisanal silica-based high-throughput methods [12,13,14,15,16].

We were interested in setting up fast, reliable, and sensitive protocols for DNA extraction from different plants for GMO detection. For such a reason, we took advantage of the ABI PRISM^®^ 6100 Nucleic Acid PrepStation (Thermo Fisher Scientific, Waltham, MA, USA), a semi-automated system used for the isolation and purification of nucleic acids, including total RNA and genomic DNA, from a variety of biological samples. The device uses standard-format 96-well plates (Genomic DNA Purification Tray I, P/N 4318641, Invitrogen™) that bind DNA through glass fiber, allowing high-throughput analyses when linked to robotic systems. According to its manufacturer, the 6100 Nucleic Acid PrepStation achieves fast (~1.5 h), high-quality isolation free of inhibitor DNA, even from complex materials including animal feed and soy flours, tofu, taco shells, and other substances with GMO content. Notably, the PrepStation benefits of the dedicated protocol used for soybean GMO detection and quantification and uses the TransPrep chemistry that, as specified in the Application Note [17], is expected to allow the detection and quantification of GMO from both reference powders of the European Commission’s Joint Research Centre for Certified Reference Materials (JRC, Geel, Belgium) and complex food. Application of the TransPrep protocol requires, besides the glass-fiber-coated Genomic DNA Purification Tray and the PrepStation itself, the use of a complete set of pre-made solutions, which are also provided by Applied Biosystems.

We compared the amplifiability of maize and soybean genomic DNA extracted with the TransPrep protocol with an EU-validated GMO quantification method [18] that refers to an official and detailed DNA extraction and purification protocol from the GM maize line, NK603, and employs CTAB, chloroform, and isopropanol [19]. In addition, we described some straightforward modifications that were exclusively applied to the buffers of the TransPrep protocol in order to increase its reliability and sensitivity, specifically those for DNA extraction, binding, and washing, used before and during the loading procedure of samples into the Genomic DNA Purification Tray loaded in the PrepStation. The appropriateness of the modifications was demonstrated using a newly designed PCR assay that targets the ribosomal 18S rRNA plant gene in order to allow quantification of both maize and soybean genomic DNAs and possibly other plant species.

## 2. Materials and Methods

### 2.1. Maize and Soybean Samples

Samples were maize and soybean dry seeds purchased from local suppliers and stored at RT. This work was initially focused on maize and soybean because they are the most common species recipients of genetic modifications and because of the availability of UE-validated DNA methods, i.e., the maize line NK603 CTAB protocol [19], that we would have needed as a reference protocol. Our experiments were based on flour because of seed availability, straightforward conservation, ease of preparation, and because most protocols are based on this matrix, i.e., the already-mentioned NK603 or the Applied Biosystems TransPrep protocols (see the next section).

### 2.2. Reagents

EBPT2 extraction buffer: 1% SDS, 1% PVP, 5% Tween 20, 0.5% Triton X-100, 50 mM EDTA, 100 mM NaCl, 100 mM Tris-HCl pH 8.RNase stock: Ribonuclease A at 20 mg/mL (Sigma Aldrich, St. Louis, MO, USA).Proteinase K stock: 20 mg/mL (Takara Bio Inc., Otsu, Japan).A total of 5 M potassium acetate stock [20].Binding Buffer BBU stock solution: 2.8 M guanidine-HCl, 65% ethanol.Wash Buffer CZ: 0.5 M guanidine-HCl, 40% ethanol.TE buffer [20].Genomic DNA Purification Trays (P/N 4318641, Invitrogen™, Carlsbad, CA, USA).

### 2.3. Instrumentations

ABI PRISM^®^ 6100 Nucleic Acid PrepStation (Thermo Fisher Scientific, Waltham, MA, USA).Refrigerated microcentrifuge.Thermomixer.

### 2.4. GUST2 DNA Extraction Protocol

In a 1.5 mL microcentrifuge tube, mix 100 mg of flour with 660 µL of EBPT2 extraction buffer.Immediately add 10 µL of Rnase A and 10 µL of proteinase K and incubate at 65 °C with shaking (900 rpm) for 60 min (for example, in an Eppendorf Thermomixer Comfort).Add 260 µL of ice-cold 5 M potassium acetate, vortex for 30 s, and incubate in ice for 10 min.Centrifuge at 8000× *g* at 4 °C for 10 min.Transfer 260 µL of the supernatant into fresh 1.5 mL tubes and add 390 µL (1.5 volumes) of Binding Buffer BBU. Mix for at least 1 min.In the meanwhile, assemble the Genomic DNA Purification Tray onto the PrepStation and pre-wet the wells with 40 µL of Wash Buffer.Load the entire resulting volume from step 5 into the well and apply 20% vacuum for 90 s.First wash: buffer CZ, 600 µL, 20% vacuum for 90 s.Second wash: 60% ethanol, 600 µL, 20% vacuum for 90 s.Drying: 90% vacuum for 30 s (this removes residual ethanol).Add 100 µL of TE buffer.Incubate with 0% vacuum for 120 s.Elute using a 20% vacuum for 120 s.

### 2.5. TransPrep Extraction Protocol

Method Two (“DNA isolation and GMO detection/quantification in IRMM reference materials”) of the TransPrep Application Note [17] was followed (Thermo Fisher Scientific, Waltham, MA, USA). Briefly, 50 mg of seed powder is incubated in 1 mL of 1X Nucleic Acid Lysis Solution (P/N 4305895 diluted 1:1 with PBS) at 99 °C for 15 min with constant shaking at 900 rpm in an Eppendorf Thermomixer Comfort. The lysate is centrifuged at 14,000 rpm for 5 min, and 200 µL of the supernatant is mixed with 100 µL of DNA Precipitation Solution 1 (P/N 4325962) and 300 µL of DNA Precipitation Solution 2 (P/N 4325964). Finally, 200 µL of the resulting lysate is purified on the PrepStation using the original TransPrep procedure: pre-wetting of wells with 40 µL of DNA Wash Solution 1 (P/N 4325958), loading of samples at 20% vacuum for 120 s, washing with 650 µL of DNA Wash Solution 1 (20%, 90 s), washing with 650 µL of DNA Wash Solution 2 (P/N 4325960) (20%, 90 s), drying (30%, 30 s), incubation with 150 µL of DNA Elution Solution 1 (P/N 4325956) (0%, 120 s), and elution (20%, 120 s).

### 2.6. NK603 CTAB Extraction Protocol

The DNA extraction method of the EU-validated protocol [19] was followed thoroughly starting from 200 mg of maize and soybean flours. Shaking was performed in an Eppendorf Thermomixer Comfort at 900 rpm.

### 2.7. Primers, Probes, and Real-Time PCR

Primer and probe sequences used in this study are listed in Table 1. Soybean Le1 and maize zSSIIb primers and TaqMan probes were those previously used by Kuribara et al. [21]. 18S rDNA primers and probes were designed using the software Primer Express version 2.0.0 (Applied Biosystems, Foster City, CA, USA). Le1 and zSSIIb TaqMan probes were labeled with the fluorescent FAM at the 5′ end and with the quench dye TAMRA attached to the 3′ end. The MGB 18S rDNA probe was labeled with FAM at the 5′ end and MGBNFQ at the 3′ end. Primers and probes were synthesized by Invitrogen and Applied Biosystems, respectively. The final PCR reaction volume was 15 µL and contained 5 µL of template DNA, 0.2 µM of each primer, 0.2 µM of each probe, and 7.5 µL of TaqMan Universal PCR Master Mix (Thermo Fisher Scientific, Waltham, MA, USA). The PCR reactions were run on an ABI Prism 7900HT Sequence Detection System device (Applied Biosystems, Foster City, CA, USA) using the following thermal protocol: 2 min at 50 °C and 10 min at 95 °C followed by 40 or 50 cycles of 15 s at 95 °C and 1 min at 60 °C, with data collection at the annealing step. PCR results were analyzed using the Sequence Detection System software version 2.2 (Applied Biosystems, Foster City, CA, USA). For all the analyses, the “threshold value” was set manually in all runs to 0.12, which matched the early stage of the linear phase of all the amplification curves.

### 2.8. Construction of the 18S rRNA Standard Reference Plasmid (p18S)

The 137 bp 18S rRNA gene fragment was amplified from 50 ng of maize genomic DNA and cloned into the pCR 2.1 TA Cloning vector (Thermo Fisher Scientific, Waltham, MA, USA) following manufacturer instructions.

## 3. Results

### 3.1. TransPrep DNA Extraction Protocol

We firstly compared the amplifiability of maize and soybean flour DNA samples prepared following the original TransPrep protocol on the PrepStation with samples prepared with the validated NK603 CTAB protocol. Samples were quantified using a NanoDrop ND-1000 spectrophotometer (NanoDrop Technologies Inc., Montchanin, DE, USA) and 10 ng/µL of stock solutions was prepared. Four twofold serial dilutions were prepared in order to set up the PCR reactions containing final genomic DNA amounts from 50 down to 3.125 ng. Figure 1a shows the amplification plots and the standard curve obtained with maize DNA extracted with the two protocols. As shown in the figure, the position of the TransPrep standard curve drawn from the template amount and the mean *C*_T_ values are 4.8 cycles shifted up with respect to that of the CTAB protocol, which corresponds to a 28-fold lower sensitivity. Moreover, the TransPrep protocol is associated with lower PCR efficiency, as indicated by the higher slope value (−3.7144 vs. −3.2249).

Even worst results were obtained for soybean flour (Figure 1b): the original TransPrep protocol yielded amplification plots with 5.8 cycles of mean delay, which means a 55-fold lower sensitivity. As shown by the amplification plots, the last dilution came out of the linearity of quantification, as depicted also by a low correlation factor value.

The TransPrep protocol is associated with frequent clogging of the Purification Tray membranes that cause irreversible obstruction of the well and sample loss. This phenomenon was especially observed when working with soybean flour but also with other food matrices and was dependent on the amount of material used in the homogenization step.

### 3.2. GUST2 DNA Extraction Protocol Evaluation Using the 18S rRNA Gene PCR Assay

The GUST2 extraction method was tested in combination with a PCR protocol that targets the ribosomal 18S rRNA plant-specific gene in order to allow quantification of both maize and soybean genomic DNAs and possibly other plant species. For such a purpose, using the ClustalW online tool [22], we found within the 18S rRNA gene a 137 bp DNA sequence that was highly conserved in plants while exhibiting weaker homology with non-plant species. The differences were weak but sufficient to design a non-plant TaqMan MGB discriminating assay (Figure 2). We exploited two highly conserved plant-specific positions, an adenine and a cytosine, in correspondence with the 3′ and the 5′ ends of the forward and reverse primers, respectively, that are substituted by other nucleotides in non-plant species. Primers and probes were tested for their plant-uniqueness: real-time PCR reactions were set up using 10,000 genome copies from some representative non-plant species, i.e., *Escherichia coli*, *Saccharomyces cerevisiae*, *Neurospora crassa*, and *Homo sapiens* gave no amplification. On the other hand, efficient amplification was obtained from genomic DNA from the plant species *Zea mays*, *Glycine max*, *Arabidopsis thaliana*, *Triticum aestivum*, and *Lycopersicon esculentum*.

The efficiency of the 18S PCR amplification assay was demonstrated by setting up in triplicate PCR reactions from four tenfold serial dilutions of a standard reference plasmid (p18S) containing the 137 bp-cloned sequence, starting from the 10^5^ plasmid copies that were spectrophotometrically calculated. As shown in Figure 3a, the quantification was linear over the range tested from 100 to 10^5^ calculated starting plasmid copies.

### 3.3. Absolute LOD and LOQ Determination

PCR reactions were set up in quadruplicate, using the 18S rRNA gene primers and probe, from twelve tenfold serial dilutions starting from 10 ng of genomic DNA, which corresponds approximately to 1835 and 4444 genome copies, respectively, for maize and soybean since one haploid genome weighs 2.73 and 1.13 pg, respectively [23]. Standard curves drawn from the DNA input and mean *C*_T_ values had slopes of −3.6655 and −3.5587, respectively, for maize and soybean (Figure 3b). Quantification was linear down to the fifth dilution, indicating an LOQ of at least 0.1 pg of maize or soybean genomic DNA per reaction, equivalent to 0.018 and 0.044 genomes per reaction, respectively (Figure 3b). Log plots of maize and soybean genomic DNA input amounts versus delta*C*_T_ calculated with respect to the p18S plasmid dilutions had a slope < of 0.1, indicating comparable amplification efficiencies. Using the standard curve established for the p18S plasmid, we thus calculated the LOQ in terms of 18S rRNA gene targets. As shown in Table 2, such amounts correspond to 34 and 109 copies. On the other hand, the lowest amount of diluted DNA that was detectable in all replicates (which corresponds to the LOD value) matched the sixth dilution, containing 0.01 pg of genomic DNA, i.e., 3.4 and 10.9 copies of 18S rRNA gene targets.

## 4. Discussion

We tested the amplifiability of genomic DNAs extracted from maize and soybean flours using the original PrepStation protocol and TransPrep reagents and found significantly lower sensitivity compared to the conventional CTAB-based extraction protocol (Figure 1). Lower sensitivity could depend on contamination with RNA, a well-known PCR inhibitor [24], as demonstrated by agarose gel electrophoresis and nuclease analysis of TransPrep eluates. Notably, by introducing a chloroform extraction step before the spectrophotometric quantification, we could increase the PCR sensitivity of the official TransPrep protocol to that of the NK603 CTAB protocol.

CTAB/chloroform-based DNA extraction protocols are generally preferred from other methods for the efficient removal of contaminants and the inhibition of nuclease activity. However, chloroform is harmful and requires supplementary handling. Hence, we decided to set up a safer and faster DNA extraction protocol to be applied to the ABI PRISM^®^ 6100 Nucleic Acid PrepStation, a 96-well-based complete system for the high-throughput extraction of nucleic acids from several different sources, equipped with a GMO quantification protocol that uses the proprietary TransPrep chemistry [17]. Our method uses an SDS-based lysis buffer (GUST2) combined with potassium acetate precipitation for the removal of contaminants and prior DNA binding to the silica with precipitation solutions. Clearing the lysis mixture and the precipitation of carbohydrates and other contaminants with a combination of SDS and potassium acetate is a well-known approach [2] and represents the principle of several commercial DNA extraction kits. The GUST2 buffer also contains Tween 20 and Triton X-100. We found that the inclusion of these detergents gave better separation in the centrifugation step and diminished the frequency of well clogging. Further modification of the official TransPrep protocol was obtained by setting up and finely tuning home-made binding and washing solutions. Such solutions contain chaotropic salts and alcohol, which are necessary for the binding of nucleic acids to glass fiber. We tested different chaotropic salts (guanidine-HCl and guanidine-isotiocianate) and alcohol (ethanol and isopropanol) at different final volumes and concentrations, and better results were obtained with guanidine-HCl and ethanol. The best combination was found to be 1.5 volumes of binding buffer, with guanidine-HCl and ethanol at the final concentrations of 1.68 M and 39%, respectively, for both maize and soybean flours. While higher concentrations of guanidine-HCl did not significantly increase the DNA yield, lower concentrations of guanidine-HCl were associated with insufficient retention of DNA in glass fiber. On the other hand, higher concentrations of ethanol (higher than 60% final concentration) caused RNA retention in the eluate and occasionally sample loss because of irreversible well clogging, a phenomenon that we observed frequently with TransPrep precipitation solutions, suggesting that these solutions may contain too high alcohol concentration.

The 18S rRNA real-time assay was designed employing an MGB probe instead of a classical TaqMan probe (Figure 2). MGB probes are dual-labeled probes conjugated with minor groove binder ligands at the 3′-end. Compared to ordinary probes, MGB probes have a higher melting temperature, which allows shorter primer design. Moreover, because of their higher specificity, MGB probes are recommended for allelic discrimination assays. Determination of the copy number by a real-time PCR requires the building of calibration curves through the quantification of calibrators such as plasmid DNA. The standard curve drawn from the p18S plasmid copy number and mean *C*_T_ values had a slope of −3.5901, indicating a PCR efficiency of 90% (Figure 3a). The efficiency was similar to that observed using maize or soybean genomic DNAs (Figure 3b). As shown in the figure, the GUST2 buffer gave a quantification linearity range of at least six orders of magnitude. Using the standard curve obtained from the p18S plasmid, we calculated 18S rRNA gene copies in our maize and soybean flour samples. Soybean 18S rDNA calculated values are consistent with those previously reported [25,26,27]. On the other hand, a smaller number of repeats were found for maize compared to previous findings [28,29,30]. However, the high rRNA gene copy number of plants was shown to change quickly without phenotypic effects [31]. The plasmid standard curve also allowed us to calculate the LOD and LOQ values in terms of target 18S rRNA gene copies rather than of input genomic DNA. Such values are in agreement with theoretical real-time PCR values [32]. The plant-universal TaqMan MGB probe increases the potential features of the detection method. Firstly, it could be used to test the quality of the genomic DNA extracted from different plant species or matrices. Second, if used in combination with a plant species-specific gene (reference gene), it could allow the relative quantification of the amount of the specific ingredient with respect to other plant ingredients. Finally, the use of a plant-specific probe could allow the determination of the GMO content relative to the total amount of vegetable tissue of the foodstuff analyzed rather than to a single species-specific ingredient.

## 5. Conclusions

The isolation of plant DNA is complicated because of the large amounts of polysaccharides, phenolics, and other metabolites in plant tissues that affect PCR sensitivity. CTAB lysis, combined with phenol/chloroform extraction and ethanol precipitation, has been successfully applied to overcome this difficulty; however, such methods are complicated and time-consuming. Taking advantage of the Genomic DNA Purification Tray (Invitrogen™, Carlsbad, CA, USA), a commercially available standard format 96-well plate that separates nucleic acids thanks to their binding to glass fiber, we used self-made buffers to set up an easy and high-throughput, compliant maize and soybean genomic DNA extraction protocol (GUST2), thanks to which we increased the low PCR sensitivity observed using the official dedicated TransPrep protocol. High PCR sensitivity and efficiency of the optimized DNA extraction protocol were demonstrated using serial dilutions of a standard reference plasmid containing a plant-specific ribosomal 18S rRNA gene sequence targeted by an expressly designed TaqMan MGB discriminating assay PCR. Altogether, the technical modifications yield a completely renovated, fast (~1.5 h), and affordable (around EUR 2 per sample) extraction protocol that substitutes the extraction, binding, and washing TransPrep solutions with favorable self-made ones, featuring on the one hand the speed of a commercial genomic DNA extraction kit and on the other hand the reliability and sensitivity of the CTAB-based EU-validated method for GMO quantification.

## Figures and Tables

**Figure 1 genes-14-01797-f001:**
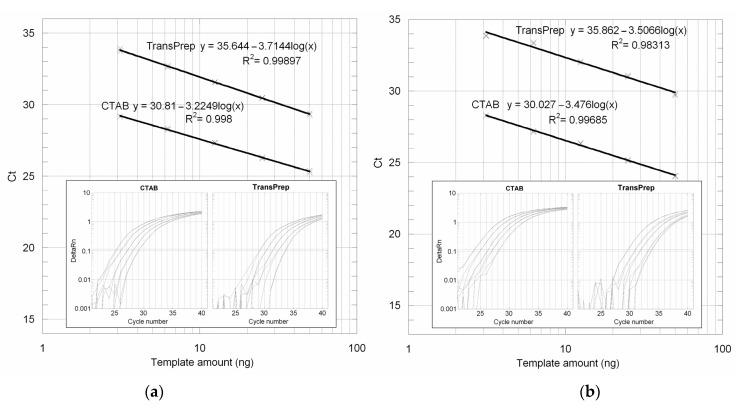
Amplification plots and standard curves obtained from maize (**a**) and soybean (**b**) flours extracted with CTAB and TransPrep protocols using, respectively, zSSIIb and Le1 primers and probes.

**Figure 2 genes-14-01797-f002:**
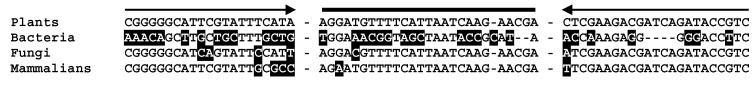
ClustalW multiple alignment of the portion of the 18S rRNA gene used for primer (arrows) and probe (line) design of the universal Plant Taqman MGB probe. Highlighted nucleotides are group-specific positions that differ from plant genes.

**Figure 3 genes-14-01797-f003:**
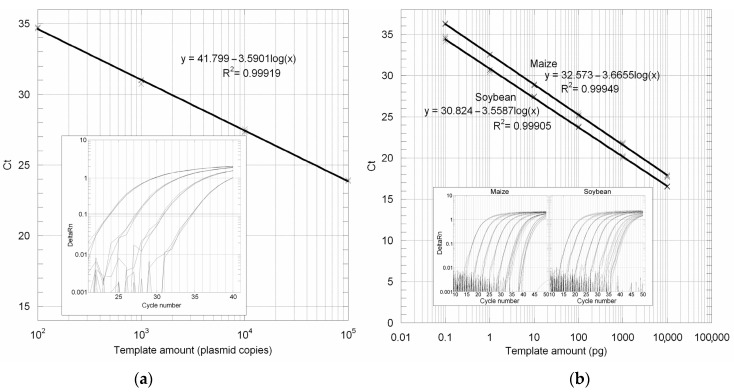
(**a**) Amplification plots and standard curve obtained by real-time PCR from plasmid p18S containing the plant 137 bp 18 sRNA conserved gene sequence. (**b**) Amplification plots and standard curves obtained by real-time PCR with the 18S rDNA TaqMan MGB primers and probe from maize and soybean genomic DNAs prepared using the extraction protocol GUST2.

**Table 1 genes-14-01797-t001:** Primers and probes used in this study.

Target (Ref)	Name	Orientation	Sequence (5′-3′)
zSSIIb [21]	SSIIb 1-5′ SSIIb 1-3′ SSIIb-Taq	forward primer reverse primer TaqMan probe	CTC CCA ATC CTT TGA CAT CTG C TCG ATT TCT CTC TTG GTG ACA GG AGC AAA GTC AGA GCG CTG CAA TGC A
Le1 [21]	Le1n02-5′ Le1n02-3′ Le1-Taq	forward primer reverse primer TaqMan probe	GCC CTC TAC TCC ACC CCC A GCC CAT CTG CAA GCC TTT TT AGC TTC GCC GCT TCC TTC AAC TTC AC
18S rDNA (this study)	uniPLA FW uniPLA RV uniPLA pro	forward primer reverse primer TaqMan MGB probe	CGG GGG CATT CGT ATT TCA TA GAC GGT ATC TGA TCG TCT TCG AG AGG ATG TTT TCA TTA ATC AAG AAC GA

**Table 2 genes-14-01797-t002:** PCR quantification of DNA dilutions of maize and soybean genomic DNAs using the 18S plasmid standard.

			Measured Values							
Calculated Values							18S rRNA Gene Copies			
DNA	Genome Copies		Mean *C*_T_ and SD				Total		per Haploid Genome	
(pg)	Maize	Soybean	Maize		Soybean		Maize	Soybean	Maize	Soybean
10,000	1835	4444	17.79	0.11	16.54	0.03	4,875,865	10,853,777	1329	1221
1000	183	444	21.80	0.10	20.15	0.10	372,780	1,073,526	1016	1208
100	18.3	44.4	25.20	0.17	23.76	0.07	41,897	106,086	1142	1193
10	1.83	4.44	28.86	0.08	27.45	0.04	4021	9899	1096	1114
1	0.18	0.44	32.52	0.05	30.54	0.36	384	1370	1048	1542
0.1	0.018	0.044	36.28	0.06	34.48	0.26	34	109	937	1231

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
