# Peer review of "Development of a Sensitive, Easy and High-Throughput Compliant Protocol for Maize and Soybean DNA Extraction and Quantitation Using a Plant-Specific Universal Taqman Minor Groove Binder Probe"

_genes, 2023, doi:10.3390/genes14091797_

Round 1

Reviewer 1 Report

The manuscript is well written and presents great relevance

Author Response

We thank the reviewer for the positive opinion

Reviewer 2 Report

Manuscript still need more work

The abstract need to rewrite again and focus on the different methods and the tangible results 

Introduction need more data on the different DNA extraction methods used for Maize and Soybean 

Overall, you wrote Maize and Soybean and some parts you changed Soybean and Maize!

I prefer to merge the results and discussion in one part to be more uncomfortable to the readers

The conclusion must rewrite again to conclude what exactly you find 

not significant 

Author Response

Point-by-point replies to reviewers’ comments:.

1) Manuscript still need more work. The abstract need to rewrite again and focus on the different methods and the tangible results.

More results and a short description of the DNA extraction procedures have been included in the abstract. We thank the referee for pointing out these deficiencies.

2) Introduction need more data on the different DNA extraction methods used for Maize and Soybean.

More information has been added on the two Maize and Soybean DNA extraction methods used to evaluate the performance of the new protocol proposed in the article, i.e. the proprietary Applied Biosystems TransPrep protocol and the EU-validated CTAB-based one.

3) Overall, you wrote Maize and Soybean and some parts you changed Soybean and Maize!

Word order has been standardized throughout the text.

4) I prefer to merge the results and discussion in one part to be more uncomfortable to the readers.

We prefer to keep Results and Discussion sections separate, because we believe that results should be reported separately (actually as required by Genes’ template) so as not to stretch them excessively, which indeed would make reading less comfortable.

5) The conclusion must rewrite again to conclude what exactly you find.

The Conclusions section has been edited in order to better focus on results.

Reviewer 3 Report

Dear Authors,

The study deals with an interesting topic, offering interesting information. The manuscript is quite well documented. The paper is well-written and I have few minor comments. I recommend the acceptance of the manuscript after revising it according to suggested minor comments. See the attached file.

Author Response

Point-by-point replies to reviewers’ comments:

1) Line 36: Why do you mention only the CTAB? Does the method work using SDS?

We mention CTAB because it is the reagent present in most conventional methods for plant genomic DNA extraction and because it is employed in a method for GM quantification validated by the EU, as mentioned in the ms.

 2) Line 303: Is the method suitable for extracting DNA from other tissues?  (i.e. animal, fungal, etc.).

The DNA extraction method was tested only with plant tissues.

3) Line 306: Is there data on cost per sample? When is the method a cost-effective solution for Labs?

Ten 96-well plates cost around 2.000€ which is around 2€ per sample, which is very competitively priced compared to magnetic based or column based kits. This indication was included in the Conclusions.

Round 2

Reviewer 2 Report

Good effort to revise the manuscript